

# Downregulated miR-383-5p contributes to the proliferation and migration of gastric cancer cells and is associated with poor prognosis

Chao Wei[1] and Jian-Jun Gao[2]

[1] Department of General Surgery, The No.967 Hospital of PLA Joint Logistics Support Force, Postgraduate Culture Base of Jinzhou Medical University, Dalian, China
[2] Department of General Surgery, The No.967 Hospital of PLA Joint Logistics Support Force, Jinzhou Medical University, Dalian, China

Corresponding author
Jian-Jun Gao, gjj1332006@126.com

## ABSTRACT

**Aim:** The study aims to identify differentially expressed microRNAs (DEMs) in gastric cancer (GC) and explore the expression, prognosis and downstream regulation role of miR-383-5p in GC.

**Methods:** The GC miRNA-Seq and clinical information were downloaded from Firebrowse which stores integrated data sourced from The Cancer Genome Atlas database. The DEMs were identified with limma package in R software at the cut-off criteria of $P < 0.05$ and |log2 fold change| > 1.0 (|log2FC| > 1.0). The expression of miR-383-5p in GC cell lines and 54 paired GC tissues was measured by quantitative real-time polymerase chain reaction (qRT-PCR). The overall survival curve of miR-383-5p and the association between its expression and clinicopathological features were explored. Wound healing and cell counting kit-8 assays were performed to investigate the capacity of miR-383-5p in cell proliferation and migration. The downstream target genes were predicted by bioinformatics tools (miRDB, TargetScan and starBase). The consensus target genes were selected for gene functional enrichment analysis by FunRich v3.0 software. The luciferase reporter assay was performed to verify the potential targeting sites of miR-383-5p on lactate dehydrogenase A (LDHA).

**Results:** A total of 21 down-regulated miRNAs (including miR-383-5p) and 202 up-regulated miRNAs were identified by analyzing GC miRNA-Seq data. Survival analysis found that patients with low miR-383-5p expression had a shorter survival time (median survival time 21.1 months) than those with high expression (46.9 months). The results of qRT-PCR indicated that miR-383-5p was downregulated in GC cell lines and tissues, which was consistent with miRNA-Seq data. The expression of miR-383-5p was significantly associated with tumor size and differentiation grade. Besides, overexpression of miR-383-5p suppressed GC cells proliferation and migration. A total of 49 common target genes of miR-383-5p were obtained by bioinformatics tools and gene functional enrichment analysis showed that these predicted genes participated in PI3K, mTOR, c-MYC, TGF-beta receptor, VEGF/VEGFR and E-cadherin signaling pathways. The data showed that expression of miR-383-5p was negatively correlated with target LDHA ($r = -0.203$). Luciferase reporter assay suggested that LDHA was a target of miR-383-5p.

**Conclusion:** The present study concluded that miR-383-5p was downregulated and may act as a tumor suppressor in GC. Furthermore, its target genes were involved in important signaling pathways. It could be a prognostic biomarker and play a vital role in exploring the molecular mechanism of GC.

# INTRODUCTION

Gastric cancer (GC) is a common malignancy of human digestive system which has high incidence and mortality worldwide (*Bray et al., 2018*). In China, both the incidence and mortality of GC rank second among malignant neoplasms (*Chen et al., 2016*). The number of Chinese patients with GC has increased over the years, which seriously threatens the health of people. The majority of Chinese GC patients are diagnosed at advanced stage and have an unsatisfactory 5-year overall survival rate (*Correa, 2013*). Most studies have found that the development of GC is associated with multiple factors, such as irregular diet, genetic and epigenetic influence (*Carcas, 2014*).

MicroRNAs (miRNAs, 20–24 nucleotides in length) are a series of non-coding RNAs and play important roles in the regulation of gene expression at post-transcriptional level (*Zhang, Wang & Gemeinhart, 2013*). Mechanistically, miRNAs negatively regulate gene expression through binding to sites in the 3′-untranslated regions of messenger RNAs (*Zen & Zhang, 2012*). Over the past years, mounting studies have confirmed that miRNAs can act as oncogenes or anti-oncogenes in the initiation and development of GC by regulating the downstream target genes (*Chen et al., 2018*; *Hui et al., 2018*; *Wang et al., 2019b*). Thus, exploring the expression and regulation role of miRNAs may be in favor of uncovering the tumorigenesis mechanism of GC.

In present study, we first identified that miR-383-5p was down-regulated in GC tissues by analyzing GC miRNA-Seq data. In order to confirm this finding, quantitative real-time polymerase chain reaction (qRT-PCR) experiment was performed to measure the expression of miR-383-5p in GC cell lines and tissues. Kaplan–Meier survival analysis also found that patients in high miR-383-5p expression group have longer overall survival time than those in low miR-383-5p expression group. Overexpression of miR-383-5p suppressed GC cells proliferation and migration. All the results showed that miR-383-5p was downregulated and it may play an anti-oncogene role in GC. The potential target genes of miR-383-5p were predicted through online bioinformatics tools. The functional enrichment analysis of target genes indicated that miR-383-5p may take part in PI3K, mTOR, c-MYC, TGF-beta receptor, VEGF/VEGFR and E-cadherin signaling pathways. LDHA, one of the 49 common target genes of miR-383-5p, was selected for target validation. The data showed that the expression of miR-383-5p was negatively correlated with LDHA ($r = -0.203$). Furthermore, a luciferase reporter assay suggested that the luciferase activity in wild-type (WT) LDHA-3′UTR group was significantly decreased by the miR-383-5p mimics, and there were no differences in the mutant LDHA-3′UTR group.

Above all, miR-383-5p can be a meaningful target in understanding the potential molecular mechanism of GC tumorigenesis and progression.

## MATERIALS AND METHODS

### MiRNA-Seq data and clinical information

The GC clinical information and miRNA-Seq data, which contain 389 cancer tissue samples and 41 gastric normal tissue samples, were downloaded from Firebrowse website (The Cancer Genome Atlas (TCGA) data version 2016_01_28). At the cut-off criterion of |log2 fold change| > 1.0 (|log2FC| > 1.0) and $P < 0.05$, the differentially expressed microRNAs (DEMs) were identified using the limma package in R software. The follow-up days and vital status of patients were extracted from clinical information data. Patients meeting the following criteria were included for overall survival: (1) patients have integrated follow-up days and vital status; (2) patients have both follow-up days and expression value of miR-383-5p. In total, 382 patients were respectively divided into low and high expression group according to the median value of miR-383-5p expression. The overall survival curve of low and high miR-383-5p expression groups were analyzed with the method of Kaplan–Meier and log-rank test.

### GC cell lines culture

Human GC cell lines (SGC-7901, BGC-823, MGC-803, and MKN-45) and a normal gastric mucous membrane cell line (GES-1) were purchased from the Institute of Biochemistry and Cell Biology of the Chinese Academy of Sciences (Shanghai, China), and all cell lines were cultivated in RPMI 1640 medium (GIBCO-BRL) supplemented with 10% fetal bovine serum (FBS; Gibco, Grand Island, NY, USA), 100 U/mL penicillin and 100 mg/mL streptomycin. All cells were cultured in humidified air at 37 °C and 5% $CO_2$.

### Patient tissues collection

All the GC tissues and the corresponding adjacent normal tissues were collected from 54 patients who received surgical resection at the No. 967 Hospital of PLA Joint Logistics Support Force and the Northern Theater Command General Hospital. Tissues were histologically confirmed and immediately stored at −80 °C after resection. The clinicopathological features of 54 GC patients were recorded and preserved. All patients signed the informed consent and this study was approved by the Research Ethics Committee of Jinzhou Medical University (IRB No: EC-2018-JZ-016).

### RNA extraction and qRT-PCR assays

The TRIzol reagent (Invitrogen, Carlsbad, CA, USA) was utilized for extracting total RNA of tissues and cells. Reverse Transcription Kit (GenePharma, Shanghai, China) was used for obtaining cDNA reverse transcribed from RNA. qRT-PCR assay was performed with SYBR-Green Hairpin-it™ MicroRNAs Kit (GenePharma, Shanghai, China), which was conducted on ABI 7500 FAST Real-Time PCR System. The expression level was

**Table 1 Sequence of primers used for PCR.**

| Name | Sequence (5′-3′) |
| --- | --- |
| miR-383-5p (RT) | GTCGTATCCAGTGCGTGTCGTGGAGTCGGCAATTGCACTGGATACG ACAGCCAC |
| miR-383-5p (forward) | GGGAGATCAGAAGGTGATTGTGGCT |
| miR-383-5p (reverse) | CAGTGCGTGTCGTGGAGT |
| U6 (forward) | CTCGCTTCGGCAGCACA |
| U6 (reverse) | AACGCTTCACGAATTTGCGT |

determined using $2^{-\Delta\Delta Ct}$ method and normalized to U6. All the sequences of primers used in present study were summarized in Table 1.

## RNA oligonucleotide and cell transfection

The miR-383-5p mimics and mimics NC were designed and synthesized by GenePharma Co., Ltd (Shanghai, China). According to the manufacturer's protocols, the GC cells were transfected using Lipofectamine™ 3000 reagent (Invitrogen, Carlsbad, CA, USA; Thermo Fisher Scientific, Waltham, MA, USA). All the cells were cultivated for 48 h after transfection.

## Cell proliferation assay

After transfection, the MGC-803 or MKN-45 cells were seeded into a 96-well plate with $4 \times 10^3$ cells per well in triplicate. At 0, 24, 48, and 72 h, each well was added with 10 μL of cell counting kit-8 (CCK-8) reagent (Dojindo, Kumamoto, Japan) and then incubated at 37 °C for 3 h. The absorbance at 450 nm was measured using a spectrophotometer.

## Wound healing assay

The MGC-803 or MKN-45 cells were seeded into six-well plate. When the cells were cultured to a density of 90%, a 100 μL pipette tip was used to draw a straight wound. Then, the cells were cultured with serum-free medium in the humidified incubator. At 0 and 48 h, an inverted microscope was utilized to visualize the wound healing and photograph. Before observing the healing status, mitomycin C (10 ug/mL) was added into each well for 2 h to exclude the influence of proliferation on cell migration.

## Target genes prediction of miR-383-5p and functional enrichment analysis

The bioinformatics websites of TargetScan (Agarwal et al., 2015) (http://www.targetscan. org/vert_71/), miRDB (Wang, 2016) (http://www.mirdb.org/) and starBase (Li et al., 2014) (http://starbase.sysu.edu.cn/) were applied for predicting potential target genes of miR-383-5p. The consensus results of the three tools were selected for further analysis. Genes functional enrichment analysis was performed by FunRich v3.0 software which was a widely used tool for the gene functional enrichment and interaction network analysis (Pathan et al., 2015). All the procedures were conducted according to official protocols and default parameters.

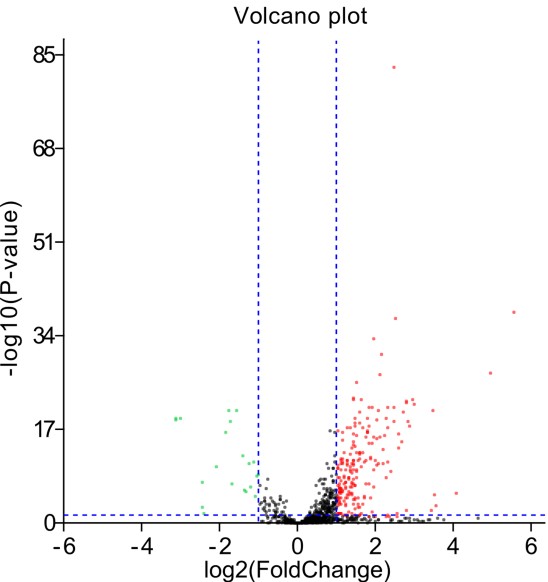

**Figure 1 The volcano plot.** The red spots represent 202 up-regulated miRNAs and the green spots represent 21 down-regulated miRNAs (blue imaginary lines represent Fold change: ±2 and *P*-value: 0.05).

## Luciferase reporter assay

The luciferase reporter vectors containing WT or mutant (MUT) 3′UTR of LDHA and miR-383-5p mimics or negative control were transfected into MKN-45 cells. Luciferase activity was tested using the Dual-Luciferase Reporter Assay System (Promega, Madison, WI, USA). Activities were normalized to Renilla luciferase. All experiments were performed three times.

## Statistical analysis

MiRNA-Seq data was processed by the limma (*Ritchie et al., 2015*) package in R software. The survival curve was described by Kaplan–Meier survival plot and analyzed with log-rank test. The differences between the two groups were analyzed by paired or unpaired Student's *t*-test. The Chi-square test was used for exploring the association between miR-383-5p expression and clinical features (such as: age, gender, tumor size, lymph node metastasis, TNM stage, and differentiation grade). $P < 0.05$ was recognized as statistically significant and all statistical analysis were conducted by IBM SPSS software 19.0.

## RESULTS

### Identification of DEMs in GC

At the cut-off criterion of |log2FC| > 1.0 and $P < 0.05$, 223 DEMs were identified by screening GC miRNA-Seq data. A volcano plot was drawn to visualize the 21 down-regulated and 202 up-regulated miRNAs (Fig. 1). The top 20 of down- and up-regulated miRNAs ranked by FC was listed in Table 2. We found that miR-383-5p was significantly down-regulated with a log2FC of −1.12, which indicated that it may act as a tumor-suppressor in GC.

**Table 2 A total of 42 DEMs identified between GC and adjacent normal tissues.**

| Down-regulated | logFC | P-value | Up-regulated | logFC | P-value |
|---|---|---|---|---|---|
| miR-1-3p | −3.13 | 1.56E-19 | miR-196a-5p | 5.55 | 3.01E-39 |
| miR-133a-3p | −3.13 | 9.66E-20 | miR-196b-5p | 4.94 | 3.33E-28 |
| miR-133b | −3.03 | 9.52E-20 | miR-767-5p | 4.05 | 3.28E-06 |
| miR-802 | −2.46 | 0.000768 | miR-552-3p | 3.55 | 0.000579 |
| miR-490-3p | −2.46 | 2.63E-08 | miR-105-5p | 3.49 | 7.65E-06 |
| miR-1265 | −2.43 | 0.014022 | miR-135b-5p | 3.46 | 2.25E-21 |
| miR-204-5p | −2.10 | 3.67E-11 | miR-767-3p | 3.44 | 0.003757 |
| miR-145-5p | −1.87 | 3.04E-17 | miR-194-5p | 2.97 | 2.20E-22 |
| miR-139-3p | −1.76 | 2.84E-21 | miR-200a-5p | 2.94 | 2.40E-23 |
| miR-145-3p | −1.75 | 2.37E-19 | miR-192-5p | 2.87 | 1.71E-18 |
| miR-129-5p | −1.70 | 4.41E-08 | miR-200a-3p | 2.81 | 2.26E-19 |
| miR-139-5p | −1.59 | 2.49E-21 | miR-200b-3p | 2.80 | 1.08E-22 |
| miR-30a-3p | −1.41 | 4.25E-13 | miR-141-5p | 2.79 | 5.05E-23 |
| miR-490-5p | −1.40 | 6.09E-07 | miR-1269a | 2.78 | 0.003274 |
| miR-551b-3p | −1.32 | 1.73E-06 | miR-183-5p | 2.71 | 4.27E-21 |
| miR-143-3p | −1.27 | 1.05E-11 | miR-194-3p | 2.64 | 1.03E-15 |
| miR-486-5p | −1.22 | 2.70E-07 | miR-429 | 2.58 | 2.15E-14 |
| miR-29c-3p | −1.14 | 7.86E-12 | miR-141-3p | 2.57 | 5.25E-17 |
| miR-383-5p | −1.12 | 1.13E-05 | miR-675-5p | 2.55 | 0.011142 |
| miR-195-3p | −1.06 | 1.83E-09 | miR-146b-5p | 2.50 | 4.41E-38 |

## miR-383-5p was confirmed to be downregulated in GC

The expression value of miR-383-5p was extracted from miRNA-Seq data. Totally, there were 387 GC samples and 41 normal samples. The expression of miR-383-5p in GC was significantly lower than normal tissues (Fig. 2A). The results of qRT-PCR showed that miR-383-5p was significantly down-regulated in GC at the level of cell and tissue (Figs. 2B and 2C). Combining the miRNA-Seq data and qRT-PCR assay, we confirmed that miR-383-5p was down-regulated and might be a novel tumor suppressor gene in GC.

## Association between miR-383-5p and prognosis, clinicopathological features

A total of 382 GC TCGA samples with necessary data were selected to investigate the prognostic role of miR-383-5p. According to the median expression value of miR-383-5p, patients were equally divided to the low and high expression groups. The Kaplan–Meier survival analysis indicated that patients with low miR-383-5p expression had a shorter survival time (median survival time 21.1 months) than those with high expression (46.9 months) (Fig. 3). Furthermore, we explored the association between miR-383-5p expression and clinicopathological features. The results manifested that low miR-383-5p expression was significantly associated with large tumor size and poor differentiation grade (Table 3). Nevertheless, the features of age, gender, lymph node metastasis, and TNM stage were found to be of no significant difference.

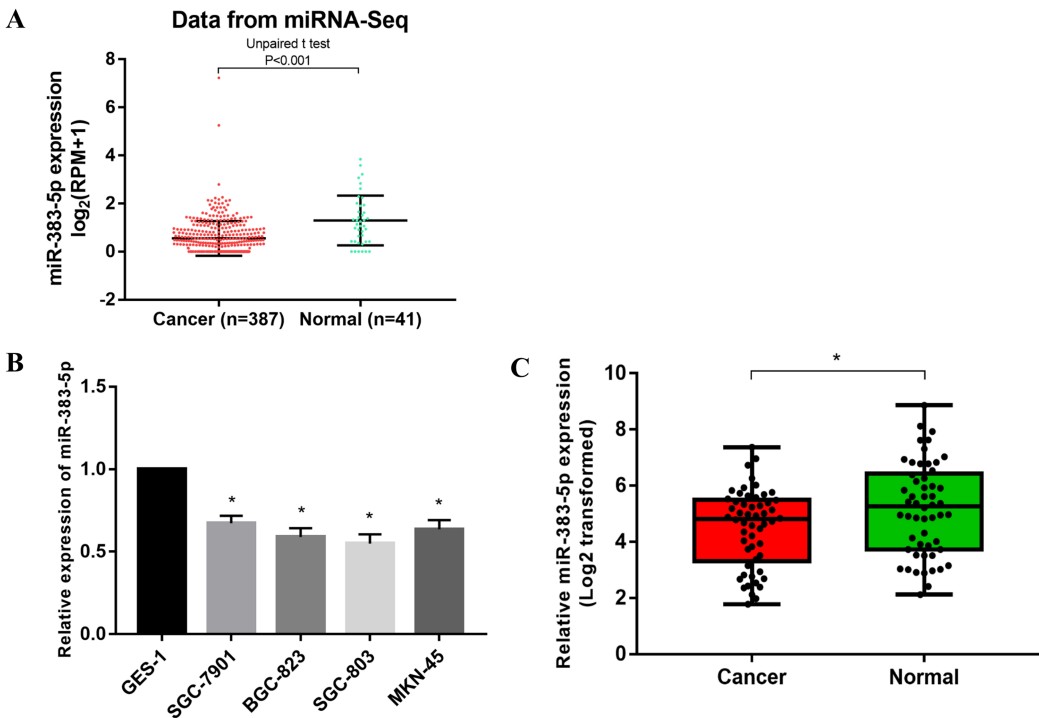

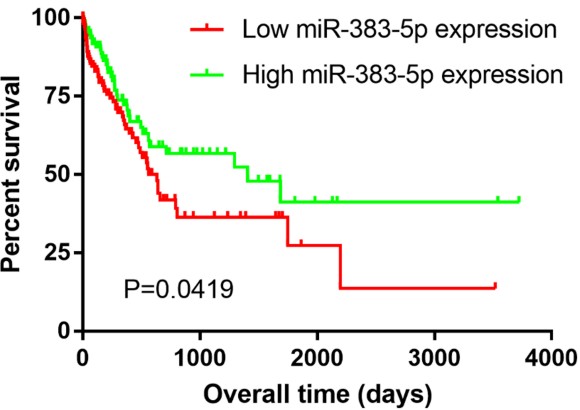

**Figure 2 The expression of miR-383-5p in GC tissues and cells.** (A) miRNA-Seq data indicated that miR-383-5p was down-regulated in GC tissues; (B) and (C) qRT-PCR showed that miR-383-5p was significantly down-regulated in GC cells and tissues. *P < 0.05 compared with the control group (GES-1 or normal tissues).

**Figure 3 The Kaplan–Meier overall survival curve.** The group with low miR-383-5p expression had a significantly less survival time (median survival time 21.1 months) than that with high expression (46.9 months).

## Overexpression of miR-383-5p inhibits GC cell proliferation and migration

The CCK-8 and wound healing assays were performed to assess the effect of miR-383-5p on the proliferation and migration of GC cells. Compared with NC group, transfection with miR-383-5p mimics weakened the migration capacity of MGC-803 and MKN-45 cells

**Table 3 Association between the genes and clinical features.**

| Variables | miR-383-5p expression | | Total samples | Pearson *r* | *P*-value |
|---|---|---|---|---|---|
| | Low (*n*, %) | High (*n*, %) | | | |
| Age | | | | | |
| <60 | 11 (17.1) | 12 (13.4) | 23 | −0.037 | 0.783 |
| ≥60 | 16 (28.0) | 15 (41.5) | 31 | | |
| Gender | | | | | |
| Male | 17 (32.9) | 13 (28.1) | 30 | 0.149 | 0.273 |
| Female | 10 (25.6) | 14 (13.4) | 24 | | |
| Tumour size | | | | | |
| ≤5 cm | 5 (9.8) | 14 (15.9) | 19 | −0.349 | 0.010* |
| >5 cm | 22 (48.8) | 13 (25.6) | 35 | | |
| Lymph node metastasis | | | | | |
| Negative | 12 (13.4) | 18 (19.5) | 30 | −0.224 | 0.100 |
| Positive | 15 (45.1) | 9 (22.0) | 24 | | |
| TNM stage | | | | | |
| I + II | 13 (28.0) | 20 (15.9) | 33 | −0.266 | 0.051 |
| III + IV | 14 (30.5) | 7 (25.6) | 21 | | |
| Differentiation grade | | | | | |
| Well and moderate | 6 (17.1) | 16 (18.3) | 22 | −0.377 | 0.006* |
| Poor | 21 (41.5) | 11 (23.2) | 32 | | |

**Note:**
* $P < 0.05$, statistically significant.

(Figs. 4A–4H). Besides, the CCK-8 assay showed that miR-383-5p mimics inhibited GC cells proliferation (Figs. 4I and 4J).

## Target prediction and genes functional enrichment analyses

Three bioinformatics websites (TargetScan, miRDB, and starBase) were selected for predicting target genes of miR-383-5p. In view of that each website had diverse bioinformatics algorithm, we took the consensus results of different predictions. As described in the venn plot, 49 consensus target genes were obtained (Fig. 5A). To comprehend the function of miR-383-5p target genes, the 49 genes were used for functional enrichment analysis by FunRich. At the aspect of biological pathway analysis, we found that these genes participated in PI3K, mTOR, c-MYC, TGF-beat receptor, VEGF/VEGFR and E-cadherin signaling pathways (Fig. 5B).

## miR-383-5p targets LDHA

LDHA is one of the 49 common target genes and it has a quite higher target prediction score. The miRNA-target prediction from starBase demonstrated that miR-383-5p had potential binding site on the 3′UTR of LDHA (Fig. 6A). The association regression analysis based on starBase project indicated that miR-383-5p expression was negatively correlated with LDHA (Fig. 6B). Furthermore, luciferase reporter assay suggested that the luciferase activity in WT LDHA-3′UTR group was significantly decreased by the

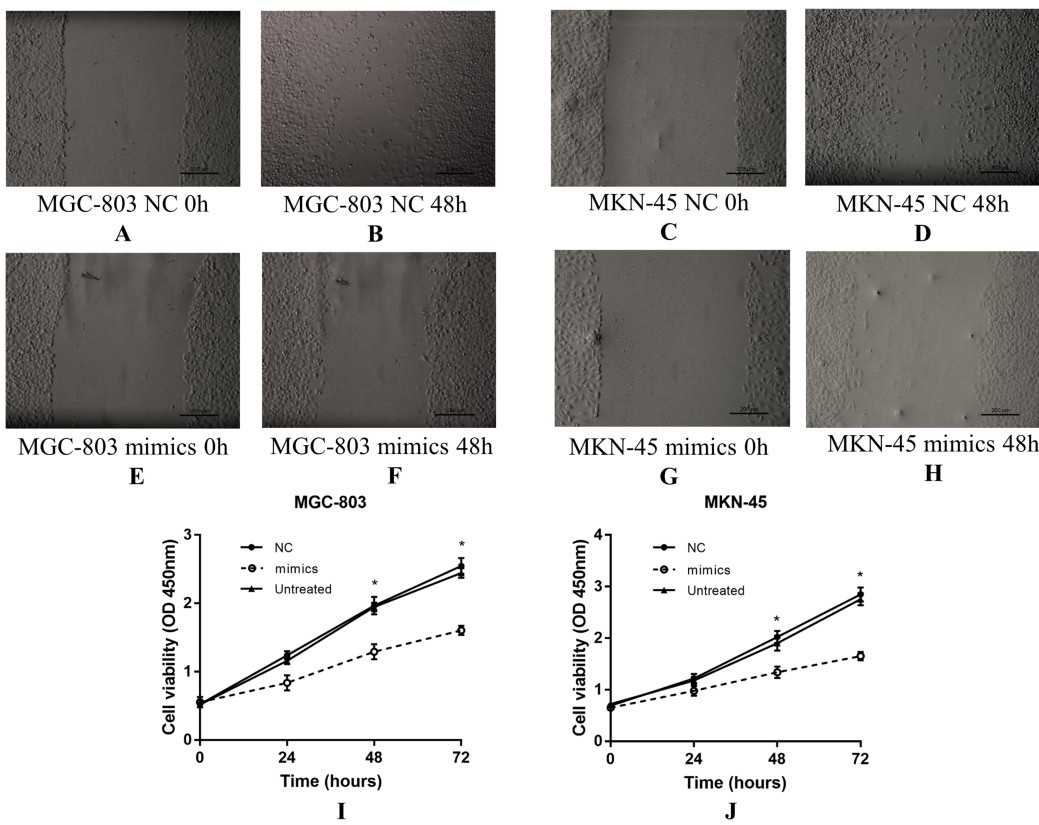

**Figure 4  The effect of miR-383-5p on the proliferation and migration of GC cells.** (A–H) miR-383-5p mimics inhibited GC cells migration; (I) and (J) miR-383-5p mimics inhibited GC cells proliferation. *$P < 0.05$ compared with NC group.

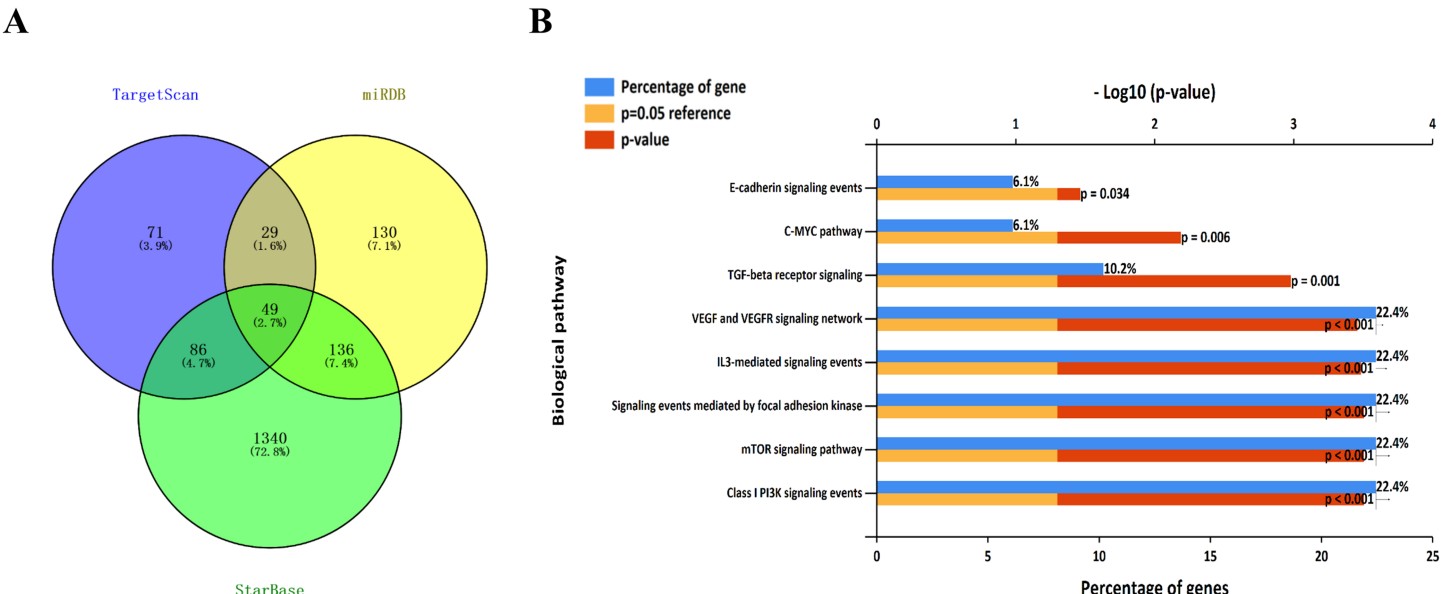

**Figure 5  The venn plot of target genes and functional enrichment analysis.** (A) A total of 49 consensus genes were obtained from TargetScan, miRDB, and starBase websites; (B) Biological pathway analysis revealed that these genes participated in PI3K, mTOR, c-MYC, TGF-beta receptor, VEGF/VEGFR and E-cadherin signaling pathways.

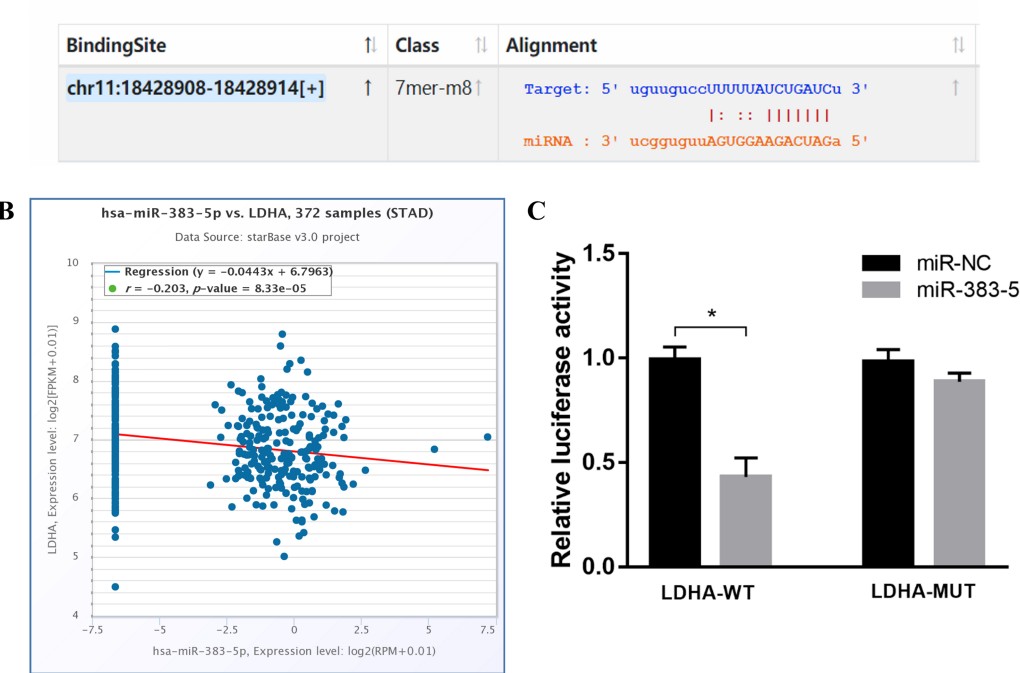

**Figure 6 miR-383-5p directly targets LDHA.** (A) The predicted miR-383-5p binding sites on the 3′-UTR of LDHA; (B) The data based on starBase project showed that expression of miR-383-5p was negatively correlated with target LDHA ($r = -0.203$); (C) Luciferase reporter assay indicated that the luciferase activity in wild-type LDHA-3′UTR group was significantly decreased by the miR-383-5p mimics. *$P < 0.05$.

miR-383-5p mimics, and there were no differences in the mutant LDHA-3′UTR group (Fig. 6C). All the results indicated that LDHA was a target of miR-383-5p.

## DISCUSSION

Gastric cancer has the malignant features of terrible proliferation, invasion, metastasis, and multiple drug resistance, which lead to high mortality and poor prognosis. Increasing studies have proved that miRNAs are aberrantly expressed and involved in the initiation and development of GC (*Chen et al., 2019*; *Kang et al., 2018*; *Maruyama et al., 2018*; *Wang et al., 2019a*). Thus, identifying DEMs and exploring the biological function of miRNAs can be useful for finding novel biomarkers and understanding the mechanism of GC progression.

In this study, we first downloaded and analyzed the GC miRNA-Seq and clinical data from Firebrowse website (*Deng et al., 2017*), which conserved integrated gene expression profiles and clinical information data from TCGA. Through screening for DEMs, we found that miR-383-5p was down-regulated in GC tissues. Besides, we also investigated the prognostic role of miR-383-5p, and Kaplan–Meier survival analysis indicated that patients with low miR-383-5p expression had a shorter survival time than those with high expression. All these results inspired us that miR-383-5p may play an important role in GC.

Furthermore, we retrieved literatures published worldwide to comprehend the studies about miR-383-5p. Zhao et al. (2017) found that miR-383-5p was significantly decreased in lung adenocarcinoma and overexpression of miR-383-5p inhibited cell proliferation by G1 cell cycle phase arrest and induced apoptosis in vitro. In hepatocellular carcinoma, miR-383-5p was proved to be a tumor suppressor and to modulate hepatocellular carcinoma tumorigenesis and progress by targeting AKR1B10 (Wang et al., 2018) and LDHA (Fang et al., 2017). Besides, Jiang et al.'s (2019) study reported that overexpression of miR-383-5p could inhibit ovarian cancer cell proliferation and enhance chemosensitivity of cells by regulating TRIM27. MiR-383-5p could also suppress ovarian cancer cell proliferation, invasion and aerobic glycolysis through regulating LDHA (Han et al., 2017). Azarbarzin et al. (2017) found that miR-383-5p was downregulated in intestinal-type GC and could be used a diagnostic biomarker. However, the molecular function and clinical significance of miR-383-5p in GC have not been studied. The present study investigated that miR-383-5p was decreased in GC and its expression was associated with tumor size and differentiation grade. Furthermore, the CCK-8 and wound healing assays demonstrated that overexpression of miR-383-5p could inhibit GC cells proliferation and migration. All the results indicated that miR-383-5p could act as a tumor suppressor in GC and it also had vital clinical value.

We further explored the downstream regulation role by predicting the potential target genes of miR-383-5p. The consensus target genes were obtained by integrating the results from three bioinformatics tools, which improved the accuracy of prediction. The functional enrichment analysis demonstrated that miR-383-5p may be involved in PI3K, mTOR, c-MYC, TGF-beta receptor, VEGF/VEGFR, and E-cadherin signaling pathways through regulating the target genes. It is well known that the mTOR pathway regulates tumor growth and metastasis by mediating tumor metabolic homeostasis (Xia & Xu, 2015). Multiple miRNAs were reported to participate in the regulation of PI3K/AKT/mTOR signaling pathway (Riquelme et al., 2016). Yu's study showed that miR-106b was overexpressed in CD44(+) GC stem-like cells and could retain cancer stem cell characteristics through modulating TGF-β/Smad signaling pathway (Yu et al., 2014). MiR-372 negatively targets KIF26B to suppresses GC cells proliferation and metastasis by regulating VEGF pathway (Zhang et al., 2017). Above all, miR-383-5p may act as a novel tumor suppressor in taking part in the biological function of GC.

LDHA acts as a glycolytic enzyme in the process of catalyzing the formation of lactate from pyruvate. LDHA is not only involved in normal cells metabolism, but closely related to tumor malignancy. Studies have found that LDHA was upregulated in kinds of malignant tumors and can play an oncogene role in GC (Wang et al., 2017; Zhu et al., 2018). The association regression analysis based on starBase project indicated that miR-383-5p expression was negatively correlated with LDHA. Besides, luciferase reporter assay showed that the luciferase activity in WT LDHA-3′UTR group was significantly decreased by the miR-383-5p mimics, and there were no differences in the mutant LDHA-3′UTR group. All the results suggested that LDHA was a downstream target of miR-383-5p.

## CONCLUSIONS

In summary, we found that miR-383-5p may act as a tumor suppressor in GC. It is of important clinical significance and prognostic value, which could contribute to revealing the molecular mechanism of GC tumorigenesis and progress.

### Funding

The authors received no funding for this work.

### Competing Interests

The authors declare that they have no competing interests.

### Author Contributions

- Chao Wei conceived and designed the experiments, performed the experiments, analyzed the data, contributed reagents/materials/analysis tools, prepared figures and/or tables, authored or reviewed drafts of the paper, approved the final draft.
- Jian-Jun Gao authored or reviewed drafts of the paper, approved the final draft, design the study.

### Human Ethics

The following information was supplied relating to ethical approvals (i.e., approving body and any reference numbers):

The Clinical Research Ethics Committee of Jinzhou Medical University approved this research (No: EC-2018-JZ-016).

### Data Availability

Data is available at Firehose, http://gdac.broadinstitute.org/: Stomach adenocarcinoma; Cohort: STAD.

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
