# Peer review of "Downregulated miR-383-5p contributes to the proliferation and migration of gastric cancer cells and is associated with poor prognosis"

_PeerJ, doi:10.7717/peerj.7882_

## Round 0.1 · original submission · Major Revisions

In addition to addressing all the points raised by both the reviewers, Please make sure that the comments made on the target genes are only based on predictions and not functionally validated. Either you can validate some of the target genes and claim the statements or tone down your conclusions that these were the predicted target genes that need replication and functional validation in experimental setting.

Reviewer 1 ·

Basic reporting

This is an interesting article describing the potential association between downregulation of miR-383-5p with the proliferation, migration, prognosis and clinicopathological features of gastric cancer.

In terms of English language used, there seems to be much room for improvement despite being submitted for language editing services. Some mistakes are listed below:
- Line 23: grammar error 'were'
- Line 50: grammar error 'mortality worldwide'
- Line 52: grammar error ' GC have increased over the years, which seriously'
- Line 55: grammar error ' with multiple factors'
- Line 79: spelling error 'clinical'
- Line 88: grammar error 'were'
- Line 95: grammar error 'in humidified'

Experimental design

The experiments used are valid in supporting the hypothesis. However, there are several comments that requires attention:

- There seems to be inconsistency between the cell line nomenclature for MGC-803 versus SGC-803.
- For wound healing assay, was a cell cycle antogonist used such as mitomycin c prior to image analysis? If so, please indicate in the methodology.

Validity of the findings

- Line 145-146: It is unclear what 'After literature retrieval' means. Please elaborate.
- What is the basis of choosing 5cm as the cut off between large vs small tumour sizes? It would be beneficial to include the standards or guidelines used in terms of such classification as variations in this measurement value can greatly influence the association and conclusions made.
- Fig 1: Why is the blue imaginary line not situated at the FC 2.0 value on the x-axis?
- Table 2: Downregulated microRNAs with FC < 2.0 are included in the Table, but Line 142 of the results mentions that FC 2.0 and p<0.05 were used as the cut-off criterion. Maintaining this cut-off for miRNA Seq data will ultimately remove miR-383-5p from the list.
- Table 2 header is wrongly labelled as Table 1.
- Table 3: P-value needs to be specified e.g. is it a comparison between low vs high groups? Based on the table, it is unclear.
- Table 3: It would be beneficial if correlation regression values be included into the table to emphasize the strength of the associations.

Additional comments

Other comments are as stated below:
- It is recommended that the term 'novel' be excluded from the manuscript as this miR-383-5p has been reported several times in past literature.

Reviewer 2 ·

Basic reporting

The manuscript is well written and easily understood. There are minor spelling errors that requires correction. (Line 79 : cliniacl – clinical; Line 100 : confrmed – confirmed; Line 102 : appvoed – approved; Line 109 : summaried – summarized; Line 187 : Firebrose – Firebrowser)

Figures are well labeled and clear. Figure 4B needs improvement. The caption is “miR-383-5p mimics inhibited GC cells proliferation,” while the y-axis in the chart uses the word absorbance. It would be better to change it to viability to make it clearer what the chart is referring to.

Experimental design

The first part of this study focuses on bioinformatic analysis on miR-383-5p expression in gastric cancer, identifying it as a downregulated miRNA in gastric cancer tissues and cell lines. The second part of the study is a functional analysis of miR-383-5p in gastric cancer through overexpression study using miRNA mimics and control in gastric cancer cell lines, followed by bioinformatic analysis on predicted target gene.

Methodology

1. Reference number/information for approval should be included in the manuscript for patient tissue collections and experiments.

2. It is unclear what is the source of the 382 GC TCGA samples (Line 85), as based on the analysis for Figure 2A, there were only 235 GC samples with available value for miR-383-5p expression. Further clarification is needed on the sample sizes and selection criteria.

Figure 2C

1. The analysis for Figure 2C is very unclear and needs to be redone. Firstly, the caption for Figure 2C is “qRT-PCR showed that miR-383-5p was significantly down-regulated in GC … tissues.” Since the 54 GC tissue samples are normalized to normal tissues, why is the normalized expression of miR-383-5p divided by half to denote low (n=27) and high (n=27) expression, when instead, the expression that is higher than normal tissue should be considered as high expression and lower as low expression.

2. Next, how was the significant difference calculated? Was the expression of miR-383-5p in GC tissue statistically compared to normal tissue for the paired t-test? In that case, it is more appropriate and accurate to have a chart comparing the expression between GC and normal tissue. The current chart appears to list the 54 GC tissue samples from low to high expression of miR-383-5p, divide the samples into two groups (low and high expression) and compare them both, which is wrong.

Table 3

It is not clear in the methodology what factors were compared to each other to calculate significant difference. Also, the total number of samples for the variable “Differentiation Grade” does not tally to 54, unlike all the other variables.

Validity of the findings

1. Although miR-383-5p overexpression study in GC cell lines is novel, its dysregulated expression in GC has been previously shown by Azarbarzin et al (Biochemical Genetics, 2017, v55(3), pp 244–252). In their study, Azarbarzin and colleagues identified miR-383-5p to be significantly downregulated in gastric cancer tissues without correlation to clinical characteristics. This highly relevant information was not discussed or addressed in this manuscript.

2. Since the overexpression of miR-383-5p inhibits migration and proliferation, and the predicted target proteins are associated with survival, authors should also look at the effects of the miRNA on apoptosis using appropriate assays instead of just observing viability. Apoptotic assays such as PARP assay or Caspase-3/7 detection assay need to be carried out to confirm the tumor suppressive effects of miR-383-5p. Knockdown of miR-383-5p is also required to support the findings of the overexpression assays.

3. Without target protein validation, it is not useful to discuss the implication of miR-383-5p relationship with the predicted signaling pathways. This is because as this point, the relationship is purely hypothetical and inconclusive. Ideally, luciferase assay and target protein RT-qPCR/western blot should be done to establish the target protein. An example can be seen in one of the papers cited in this manuscript (Zhao et al. 2017). Therefore, target validation is important before the findings of this manuscript can be published.

4. The conclusion in the abstract states “its target genes were involved in important signaling pathway.” As the target genes were only predicted and not confirmed by validation, it would be incorrect to conclude as such. This should be written as “its predicted target genes…” instead unless validation is carried out. Furthermore, this is different compared to the conclusion at the end of the manuscript, which does not make any reference to the target genes.

---

## Round 0.2 · accepted · Accept

Based on the external review and internal evaluation it was felt that the manuscript quality has been improved and all the points raised by the reviewers were addressed.

Reviewer 1 ·

Basic reporting

No further comment.

Experimental design

The authors have addressed all comments raised.

Validity of the findings

No comment.

Additional comments

The manuscript has been significantly improved, and the authors have addressed all comments raised.